# Sensitivity Analysis of Anchored Slopes under Water Level Fluctuations: A Case Study of Cangjiang Bridge—Yingpan Slope in China

**Jinxi Liang and Wanghua Sui ***

School of Resources and Geosciences, China University of Mining and Technology, Xuzhou 221116, China; liangjinxi@cumt.edu.cn

*   Correspondence: suiwanghua@cumt.edu.cn

**Abstract:** This paper presents an improved slope stability sensitivity analysis (ISSSA) model that takes anchoring factors into consideration in umbrella-anchored sand and clay slopes under reservoir water level fluctuation. The results of the ISSSA model show that the slope inclination and the layout density of anchors are the main controlling factors for sand slope stability under fluctuation of the water level, while the slope inclination and water head height are the main controlling factors for slope stability in the Cangjiang bridge—Yingpan slope of Yunnan province in China. Moreover, there is an optimum anchorage angle, in the range of 25–45 degrees, which has the greatest influence on slope stability. The fluctuation of the reservoir water level is an important factor that triggers slope instability; in particular, a sudden drop in the surface water level can easily lead to landslides; therefore, corresponding measures should be implemented in a timely manner in order to mitigate landslide disasters.

**Keywords:** stability sensitivity analysis; water level fluctuation; umbrella anchor



## 1. Introduction

The fluctuation of reservoir water level, which is caused by heavy rainfall, water storage, and flood discharge, may result in the instability of slopes in hydraulic engineering. In 1959, a large-area landslide occurred upstream of the Zhexi reservoir in Hunan province, which was mainly triggered by the rapid rise of the water level [1]. In October 1963, a major landslide occurred on the left bank of the Vajont reservoir in Italy, with a total volume of 240 million cubic meters, and 2600 people were killed [2]. Since 2008, the water level of the Three Gorges Reservoir area in China has been tentatively impounded by 175 m. The reservoir experiences nearly 30 m of water level fluctuation, which has caused nearly 500 ancient landslides to be revived, and the deformation and instability have led to the occurrence of new landslides [3]. Scholars and engineers have studied the causes, mechanisms, and characteristics of landslides, such as the Woshaxi, Qianjiangping, Shuping, and Quchi landslides, in the Three Gorges Reservoir area [4–8]. It is widely considered that hydraulic factors such as atmospheric precipitation, fluctuation, and the change rate of the water level of the reservoir, as well as river wave erosion, etc., could reduce the stability of bank slopes, even leading to landslide disasters. Landslides often occur after long rainy periods due to the increase in the positive pore pressure on a potential sliding surface and the decrease in the stability of deeper slopes [9–11]. If intense rainfall occurs or the water level of the reservoir fluctuates rapidly, shallow landslides can occur as the moisture content in the soil becomes close to saturation, resulting in a considerable reduction in soil strength [12,13]. According to the scale model based on the Liangshuijing landslide, slope failure mainly occurs in the drainage stage, and the stability of the reservoir bank slope would gradually deteriorate during the long-term cyclic operation of the Three Gorges Reservoir [14]. Rainfall before an earthquake will increase the water content on

the surface of the slope and reduce the strength of loess, causing a large displacement and deep cracks in the slope [15]. In general, an earthquake causes permanent damage to the slope and secondary damage results from rainstorms after the earthquake [16]. The rock slopes formed by the bedding structures of fractured rocks disturbed by human activities and subjected to heavy rainfall are liable to a loss of stability [17]. The damage of high bedding rock slopes by flood discharge of atomized rain was studied based on the slope in the Baihetan Hydropower Station. The results showed that the failure modes of the bedding rock slope were of two types: sliding–fracturing (the first slip block) and fracturing–sliding (other blocks) [18]. The long-term effects of rainfall recharge and the fluctuation of groundwater weaken the rock and soil, causing creep deformation and even landslides [19]. Sui and Zheng [20] studied the failure mode of soil slopes caused by drawdowns through transparent soil testing, dividing the process of the destabilization of coast slopes into two stages: surface sliding and overall sliding.

Not only hydrological factors but other factors, such as the properties of the slope itself (e.g., the inclination of the slope, the density, and cohesion of slope soil, etc.), also affect the stability of a slope [21]. The key variables for slope stability and the thresholds of water can be determined by sensitivity analysis, which can help engineers to identify the key variables triggering landslides.

Sensitivity analysis involves studying the influence of factor changes on the outputs by changing the values of related factors. The purpose is to identify the sensitive factors that have an important influence on the outputs from many uncertain factors and to obtain the degree of influence of different factors on the outputs. Shallow landslides with a small thickness (generally less than 2 m) are easily induced by rainfall because frequent rainfall events determine the progressive infiltration of the rainwater to the deepest soil levels [22]. It was found that the depth of soil and the inclination of the slope are the main controlling factors of slope stability, and vegetation roots can improve the stability of a slope [23]. An infinite slope stability analysis (ISSA) model, involving some factors of shallow landslides, was established by Cross [24]. The stability of shallow slopes is highly sensitive to the effective cohesion force ($c$), piezometer height ($h$), inclination of the slope ($\beta$), and average sensitivity to soil depth ($z$), and it is relatively insensitive to the effective angle of internal friction ($\varphi$) and soil unit weight ($\gamma$). The process sensitivity index, which was proposed by combining averaging methods and a variance-based global sensitivity analysis, was used to solve the problem; the model consists of multiple process-level sub-models [25]. An ISS (infinite slope stability) model applied by Choo et al. [26] showed that slope inclination and soil depth have the greatest influence on the output of the infinite slope stability model. Some researchers adopted the stacking ensemble technique, combining a radial basis function (RBF) with the random subspace (RSS), attribute selected classifier (ASC), support vector machine (SVM), least squares support vector machine (LS-SVM) [27], cascade generalization (CG) [28,29], artificial neural network (ANN), gradient-boosting decision tree (GBDT) [30], recurrent neural network (RNN), convolutional neural network (CNN) [31], and other machine learning methods. Dai et al. [32] integrated a hierarchical uncertainty framework with a variance-based global sensitivity analysis to deal with a large number of input factors, such as slope, aspect, elevation, curvature, slope length, and valley depth. Many other factors were used to develop the ensemble model's package landslide samples [33–35]. The seismic pseudo-static stability of a rock wedge was studied based on the nonlinear Barton–Bandis criterion. The sensitivity analysis showed that the parameters related to the strike of the joint plane have a greater influence than others [36].

Since water level fluctuation is a critical trigger factor for a landslide, some slope stabilization techniques can be used to alleviate this problem. The controlling techniques for landslide prevention engineering include anti-slide pile and wall systems, surface and underground drainage systems, and prestress systems [37]. Stabilizing piles are widely used for increasing the safety conditions of thrust-type slopes, which mainly control the deformation and failure process of the upper sliding mass. A stabilizing pile is a passive reinforcement measure [38]. Based on the results of a model test and field test on stabilizing

piles in China, the distribution functions of the landslide-thrust and the resistance of soil or rock have been deduced [39]. Troncone et al. combined the water balance equation with the motion equation to evaluate and predict the rainfall-induced movements of landslides in the presence of stabilizing piles [40]. Hu et al. established an in situ multi-field information monitoring platform to monitor the multi-field dynamic process of the Majiagou landslide with stabilizing piles. The monitoring results showed that the deformation of the experimental piles was influenced by the reservoir operation [41]. Current drainage systems can be divided into surface water drainage systems and subsurface drainage systems [42]. The drainage tunnel can control the decrease in the depth of the groundwater table and reduce the pore water pressure of landslides to promote the stability of slopes [37,43]. Based on the protection scheme of drainage, the groundwater level within zone II of the Jinpingzi landslide was lowered by 3.0–12.3 m, along with a decrease in the annual increment of the deformation in the same period [44]. An anchor is a common prestress technology. It can be divided into the free section and the anchoring section. The free section refers to the area where the tension at the anchor head is transmitted to the soil. By prestressing the landslide, the anti-sliding force is increased, and the sliding force is reduced [37]. According to the type of anchorage section, the anchor can be divided into a cylinder anchor and an expanded head anchor. An expanded head anchor does not need specific tools to expand the bottom and causes little disturbance to soil [45]. The main influencing factors of the end-bearing force of the expander section are the buried depth of the expanded head ($h$), the effective cohesion force ($c$), and the internal friction angle ($\phi$) [46]. An umbrella anchor is a new type of expanded head anchor. It was proposed by Smith in 1966, and its self-expanding property remarkably improved the uplift bearing capacity [47,48]. The sensitivity analysis of an umbrella-anchored rock slope under water storage conditions showed that the order of the sensitivity on the uplift-bearing capacity is the inclination of strata, amount of umbrella anchors, and water content [49]. These results are useful to evaluate the landslide sensitivity in a certain area, which can guide landslide disaster prevention.

In the study of the influence of hydraulic factors on slope stability, the fluctuations of the reservoir water or the fluctuations of the groundwater level are monitored to evaluate the slope stability. Viratjandr and Michalowski [50] proposed a model of slope stability that considers the effect of reservoir water level and groundwater level, which can be used for sensitivity analysis. Few researchers have considered anchoring factors in the analysis of slope stability, especially the umbrella anchor. This study aimed to conduct a sensitivity analysis on the stability of umbrella-anchored sand and clay slopes under the fluctuation of surface and groundwater levels.

## 2. Methods

Landslides are caused by a complex array of factors. To assess the sensitivity of various factors to sand and/or clay slopes, it is necessary to build models that consist of factors affecting slope stability.

According to the stability formula of cohesionless and dry slopes:

$$F = \frac{W \cos \beta \tan \varphi}{W \sin \beta} = \frac{\tan \varphi}{\tan \beta} \tag{1}$$

where $F$ is the factor of the safety of the slope, $W$ is the weight of the slip body (N), $\beta$ is the slope gradient (°), and $\varphi$ is the internal friction angle of the soil (°).

An improved slope stability sensitivity analysis (ISSSA) model that takes anchoring factors into consideration in umbrella-anchored slopes was presented based on a model

(Equation (2)) proposed by Viratjandr and Michalowski [50], reflecting the factor of safety of a sand slope under water level fluctuation.

$$F = \tan\varphi \frac{\frac{\gamma_\omega}{\gamma}\left(1 - \frac{L_1}{H}\right)^2 \sin^2\beta - \frac{\gamma_\omega}{\gamma}\left(1 - \frac{L_2}{H}\right)^2 + \cos^2\beta}{\left[1 - \frac{\gamma_\omega}{\gamma}\left(1 - \frac{L_1}{H}\right)^2\right]\sin\beta\cos\beta} \tag{2}$$

where $\gamma$ is the weight of soil (kN/m$^3$); $\gamma_\omega$ is the weight of water (kN/m$^3$); $H$ is the slope height (m); $\beta$ is the inclination of the slope (°); and and $L_1$ and $L_2$ are the levels of water in the reservoir and the slope measured from the crown level (m), respectively.

The latter part, except for $\tan\varphi$ in Equation (2), could be regarded as a coefficient of multi-factor slope safety, which is related to the fluctuation of the water level (Figure 1). $\theta$ is the assumed angle of failure surface.

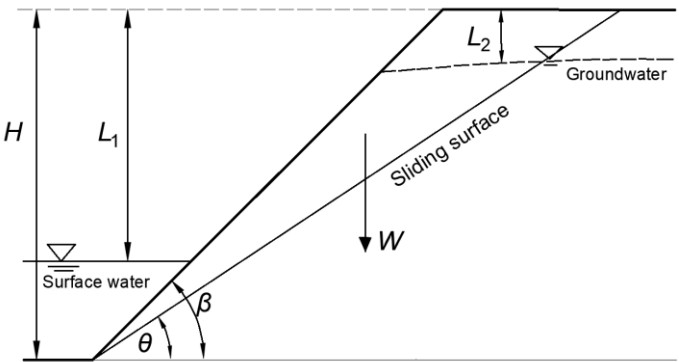

**Figure 1.** Schematic diagram of slope.

$$F = \tan\varphi \cdot \frac{W\cos\beta}{W\sin\beta} \cdot \frac{\frac{\gamma_\omega}{\gamma}\left(1 - \frac{L_1}{H}\right)^2 \sin^2\beta - \frac{\gamma_\omega}{\gamma}\left(1 - \frac{L_2}{H}\right)^2 + \cos^2\beta}{\left[1 - \frac{\gamma_\omega}{\gamma}\left(1 - \frac{L_1}{H}\right)^2\right]\cos^2\beta} = \frac{\text{Sum of restraining forces}}{\text{Sum of disturbing forces}} \tag{3}$$

Then, we can add anchoring force $T$, and assume that the angle between the anchoring force and the normal line of slope surface is $\alpha$:

$$F = \tan\varphi \frac{W\cos\beta\left[\frac{\gamma_\omega}{\gamma}\left(1 - \frac{L_1}{H}\right)^2 \sin^2\beta - \frac{\gamma_\omega}{\gamma}\left(1 - \frac{L_2}{H}\right)^2 + \cos^2\beta\right] + T\cos\alpha}{W\sin\beta\left[1 - \frac{\gamma_\omega}{\gamma}\left(1 - \frac{L_1}{H}\right)^2\right]\cos^2\beta - T\sin\alpha} \tag{4}$$

$$W = \frac{\gamma H^2 \sin(\beta - \theta)}{2\sin\theta\sin\beta} \quad (\varphi \geq \beta > \theta > 0) \tag{5}$$

The uplift-bearing capacity of the umbrella anchor can be divided into three parts: the side friction resistance between the ordinary anchorage section and surrounding soil $R_1$, the side friction resistance between the expansion section and surrounding soil $R_2$, and the end-bearing force of the expanded section $R_3$ [51]. In other words, the uplift-bearing capacity can be expressed as:

$$T = R_1 + R_2 + R_3 \tag{6}$$

The umbrella anchor's $R_3$ is much larger than its $R_1$ and $R_2$, so only the $R_3$ is considered [51].

$$T = R_3 = A\sigma \tag{7}$$

$$\sigma = \frac{(1 - \xi)K_0 K_p \rho h + 2c\sqrt{K_p}}{1 - \xi K_p} \tag{8}$$

where $A$ is the projected area of the anchor along the direction of force (m$^2$), $\sigma$ is the positive pressure strength of soil acting on the expansion end (kPa), $\xi$ is the coefficient of lateral pressure, which is equivalent to Rankine's active earth pressure coefficient $K_a$, $\xi = (0.5–0.95)$ $K_a$, $K_0$ is the static earth pressure coefficient of soil around the fixed section, $K_p$ is Rankine's passive earth pressure coefficient of soil, $h$ is the buried depth of umbrella anchor (m), $\rho$ is the density of soil (kg/m$^3$), and $c$ is the cohesion of soil (kPa).

## 3. Results and Analysis

### 3.1. Sensitivity Analysis of Anchored Sand Slope

Case 1 for the sensitivity analysis condition is a sand slope with a height of 20 m and a groundwater level of 2 m away from the crown level of the slope. Meanwhile, the surface water level is 3 m higher than the bottom of the slope, and the slope inclination is 30° (Figure 2). In other words, the height of the slope $H$ is 20 m; $L_1$ is 17 m; $L_2$ is 2 m; $\beta$ is 30°; and the internal friction angle $\varphi$ is 35°. Figure 2. Sand slope model: (**a**) sand slope section; (**b**) sand slope model with umbrella anchors.

Table 1 lists the parameters of the umbrella anchor.

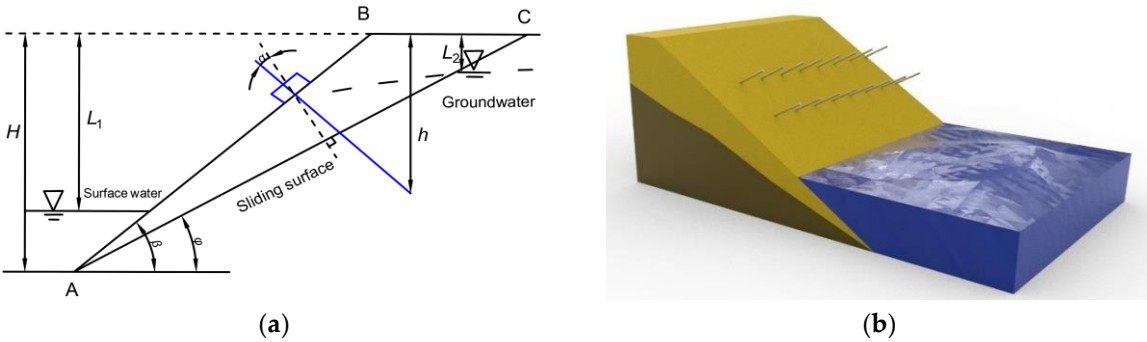

(**a**)                                                     (**b**)

**Figure 2.** Sand slope model: (**a**) sand slope section; (**b**) sand slope model with umbrella anchors.

**Table 1.** Parameters of the umbrella anchor.

| Parameter | Symbol | Value |
|---|---|---|
| The area of the umbrella anchor projected along the direction of the force | $A$ | 375 cm$^2$ |
| Static earth pressure coefficient of the soil before the anchoring section | $K_0$ | 0.25 |
| Rankine's passive earth pressure coefficient | $K_p$ | 3.7 |
| The cohesion of soil | $c$ | 0 kPa |
| The density of soil | $\rho$ | 1.9 g/cm$^3$ |
| Coefficient of lateral pressure | $\xi$ | 0.22 |

$\xi$ is equivalent to Rankine's active earth pressure coefficient $Ka$, $\xi = (0.5–0.95)$ $K_a$

According to Equations (7) and (8) for the pull-out bearing capacity of umbrella anchors, when the objective conditions are determined, the main factor affecting the pull-out bearing capacity is the buried depth of the umbrella anchor $h$. Therefore, $h$ can be regarded as the average of multiple anchors to simplify the calculation. The anchoring force $T$ is $xA\sigma$, and $x$ is the total number of umbrella anchors. The arrangement of umbrella anchors is temporarily ignored. Assuming that the sliding surface angle $\theta$ is 25°, $T \approx 0.1755W$, and $x$ is adjusted according to the weight of the sliding mass.

Equations (4), (5), (7), and (8) are the stability equations of umbrella-anchored sand slopes under water level fluctuations. The parameters involved in the formulas are determined as follows in combination with practical experience and reference values of the engineering survey. The weight of sand soil is generally 18–20 kN/m$^3$, with a reference value of 19 kN/m$^3$. The internal friction angle of sand soil is generally 20–40°, with a reference value of 35°. The related parameters of the umbrella anchor involved in this sensitivity analysis are the normal angle between the umbrella anchor and the normal line of slope surface, as well as the average buried depth of the umbrella anchor. The anchorage

angle $\alpha$ is generally 15–35°, with a reference value of 25°. The average buried depth $h$ is 4 m. The factor of safety is 1.05.

### 3.1.1. Sensitivity of a Single Factor

Sensitivity analysis is carried out on soil weight ($\gamma$), the distance between surface water level and the crown level of the slope ($L_1$), the distance between groundwater level and the crown level of the slope ($L_2$), the internal friction angle of soil ($\varphi$), slope inclination ($\beta$), the anchorage angle of the umbrella anchor ($\alpha$), and the average buried depth of the umbrella anchor ($h$). Table 2 shows the range of each factor.

**Table 2.** The variation range of each factor.

| Factor | Symbol | Reference Value | Range |
| --- | --- | --- | --- |
| Soil weight | $\gamma$ | 19 kN/m³ | 16–22 kN/m³ |
| Distance from surface water level to the crown level of the slope | $L_1$ | 17 m | 14–20 m |
| Distance from groundwater level to the crown level of the slope | $L_2$ | 2 m | 0–6 m |
| Internal friction angle | $\varphi$ | 35° | 20–40° |
| The inclination of the slope | $\beta$ | 30° | 26–34° |
| Anchorage angle | $\alpha$ | 25° | 16–34° |
| Average buried depth | $h$ | 4 m | 1–7 m |

Figure 3 shows that the stability of the sand slope is negatively correlated with $\beta$ and the factor of $L_1$ and positively correlated with other factors. Figure 4 shows that the water level fluctuation factors $L_1$ and $L_2$ have exactly the opposite effects on the factor of safety. This implies that landslides will be triggered easily with a large amount of precipitation in a short time, because this results in a sudden change in the surface water level, while the groundwater level changes lag, depending mainly on the hydraulic conductivity of the soil [52]. The fluctuation of the water level is caused by impoundment or discharge in a short time. This is similar to the fact that the minimum factor of safety occurs during the rapid declination period of the reservoir water level and rainy season [53]. When the surface water level is more than one quarter of the slope height, a sudden drop in the surface water level is more likely to trigger instability [54,55]. If the surface water level rises suddenly, the factor of safety will increase first and then decrease when the surface water level is less than 10% of the slope height. The water levels have the greatest impact on the factor of safety when the surface water level is 20–30% of the slope height. This result is close to that of Shi and Zheng [56].

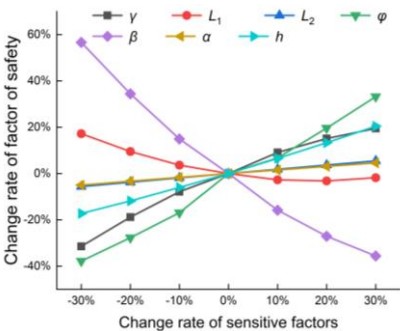

**Figure 3.** Comprehensive graph of the single factor sensitivity analysis.

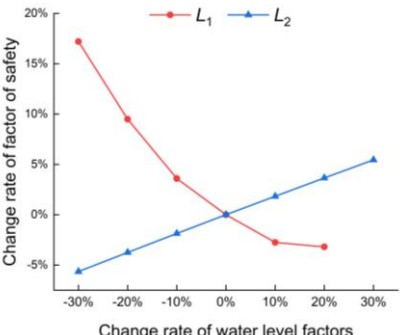

**Figure 4.** Analysis and comparison chart of water level fluctuation factors.

### 3.1.2. Sensitivity of Multiple Factors

Table 3 lists the results of the sensitivity analysis of multiple factors with an orthogonal trial $L_{25}(5^6)$. Table 4 lists the range analysis and Figure 5 shows the relationship between each factor and the stability trend.

$$R_j = \max\left(\overline{K}_{1j}, \overline{K}_{2j}, \ldots, \overline{K}_{sj}\right) - \min\left(\overline{K}_{1j}, \overline{K}_{2j}, \ldots, \overline{K}_{sj}\right) \tag{9}$$

where $R_j$ represents the degree of influence of the factor on the trial index.

**Table 3.** Results of the orthogonal experiment of sand slope stability under water level fluctuations.

| Trial | $\gamma$ (kN/m³) | $L_1$ (m) | $L_2$ (m) | $\varphi$ (°) | $\beta$ (°) | $h$ (m) | $F$ |
|-------|------|------|------|------|------|------|------|
| 1 | 16.0 | 14.0 | 0.0 | 20.0 | 26 | 1.0 | 0.70 |
| 2 | 16.0 | 15.5 | 1.5 | 24.5 | 28 | 2.5 | 0.81 |
| 3 | 16.0 | 17.0 | 3.0 | 29.0 | 30 | 4.0 | 1.00 |
| 4 | 16.0 | 18.5 | 4.5 | 33.5 | 32 | 5.5 | 1.24 |
| 5 | 16.0 | 20.0 | 6.0 | 38.0 | 34 | 7.0 | 1.54 |
| 6 | 17.5 | 14.0 | 1.5 | 29.0 | 32 | 7.0 | 1.02 |
| 7 | 17.5 | 15.5 | 3.0 | 33.5 | 34 | 1.0 | 0.53 |
| 8 | 17.5 | 17.0 | 4.5 | 38.0 | 26 | 2.5 | 4.50 |
| 9 | 17.5 | 18.5 | 6.0 | 20.0 | 28 | 4.0 | 1.11 |
| 10 | 17.5 | 20.0 | 0.0 | 24.5 | 30 | 5.5 | 0.74 |
| 11 | 19.0 | 14.0 | 3.0 | 38.0 | 28 | 5.5 | 2.88 |
| 12 | 19.0 | 15.5 | 4.5 | 20.0 | 30 | 7.0 | 1.02 |
| 13 | 19.0 | 17.0 | 6.0 | 24.5 | 32 | 1.0 | 0.53 |
| 14 | 19.0 | 18.5 | 0.0 | 29.0 | 34 | 2.5 | 0.31 |
| 15 | 19.0 | 20.0 | 1.5 | 33.5 | 26 | 4.0 | 4.00 |
| 16 | 20.5 | 14.0 | 4.5 | 24.5 | 34 | 4.0 | 0.58 |
| 17 | 20.5 | 15.5 | 6.0 | 29.0 | 26 | 5.5 | 5.00 |
| 18 | 20.5 | 17.0 | 0.0 | 33.5 | 28 | 7.0 | 2.37 |
| 19 | 20.5 | 18.5 | 3.0 | 38.0 | 30 | 1.0 | 0.78 |
| 20 | 20.5 | 20.0 | 1.5 | 20.0 | 32 | 2.5 | 0.31 |
| 21 | 22.0 | 14.0 | 6.0 | 33.5 | 30 | 2.5 | 1.09 |
| 22 | 22.0 | 15.5 | 0.0 | 38.0 | 32 | 4.0 | 0.67 |
| 23 | 22.0 | 17.0 | 1.5 | 20.0 | 34 | 5.5 | 0.36 |
| 24 | 22.0 | 18.5 | 3.0 | 24.5 | 26 | 7.0 | 6.50 |
| 25 | 22.0 | 20.0 | 4.5 | 29.0 | 28 | 1.0 | 0.75 |

**Table 4.** Range analysis for the main effects on the stability of a sand slope under water level fluctuation.

| Factor | 1 | 2 | 3 | 4 | 5 | 6 |
| | $\gamma$ | $L_1$ | $L_2$ | $\varphi$ | $\beta$ | $h$ |
|---|---|---|---|---|---|---|
| $K_{1j}$ | 1.06 | 1.25 | 0.96 | 0.7 | 4.14 | 0.66 |
| $K_{2j}$ | 1.58 | 1.61 | 1.39 | 1.83 | 1.58 | 1.4 |
| $K_{3j}$ | 1.74 | 1.75 | 2.24 | 1.62 | 0.93 | 1.47 |
| $K_{4j}$ | 1.81 | 1.99 | 1.62 | 1.85 | 0.75 | 2.04 |
| $K_{5j}$ | 1.87 | 1.47 | 1.85 | 2.07 | 0.66 | 2.49 |
| $R_j$ | 0.81 | 0.74 | 1.28 | 1.37 | 3.48 | 1.83 |
| Rank | 5 | 6 | 4 | 3 | 1 | 2 |

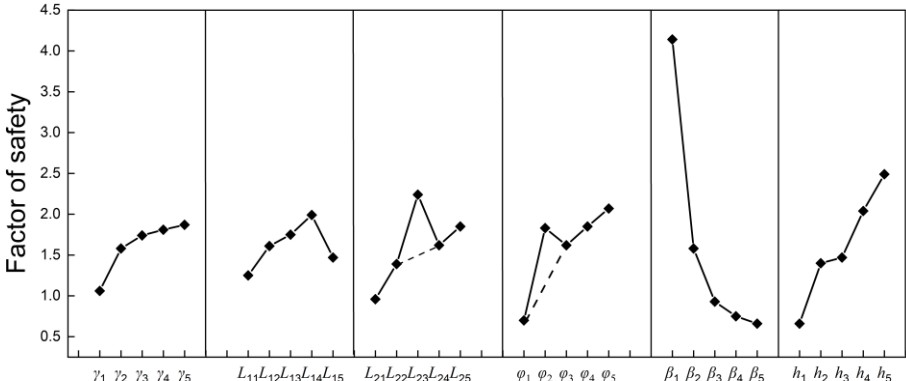

**Figure 5.** Response graphs for the main effects according to Table 4.

Table 5 shows that the order of sensitivity of each factor is as follows: $\beta > h > \gamma > \varphi > L_2 > L_1$. The factors of water level fluctuation and anchor arrangement are mainly considered. We change the slope inclination from 30° to 34°, keep the other parameters consistent (Figure 6), and analyze the sensitivity of the factors of the umbrella anchor layout and water level fluctuation. According to the specification of the layout of the anchor, the horizontal spacing of the anchor layout should not be less than 2 m. Therefore, the horizontal spacing of the umbrella anchor layout is selected as 2 m in this analysis, and *n* is the number of umbrella anchor rows, which can be understood as the number of umbrella anchors arranged in the vertical direction of slope within the unit width (2 m). Table 6 shows the variation range of each factor.

**Table 5.** Results of the variance analysis of sand slope stability.

| Source | Sum Sq. | Mean Sq. | Rank |
|---|---|---|---|
| $\gamma$ | 117.80 | 29.45 | 3 |
| $L_1$ | 87.14 | 21.79 | 6 |
| $L_2$ | 104.47 | 26.12 | 4 |
| $\varphi$ | 102.54 | 25.63 | 5 |
| $\beta$ | 810.61 | 202.65 | 1 |
| $h$ | 132.65 | 33.16 | 2 |
| Total | 1357.58 | | |

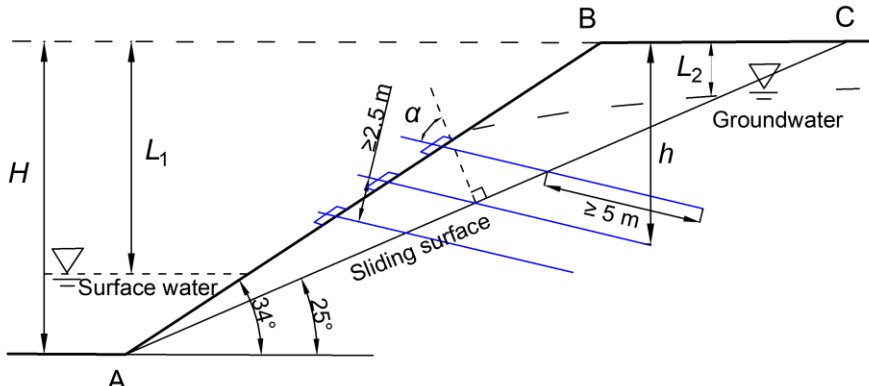

**Figure 6.** Sand slope model diagram considering the factors of anchorage.

**Table 6.** The variation range of parameters.

| Factor | Symbol | Range |
|---|---|---|
| Layout density | $n$ | 1–2 rows |
| Anchorage angle | $\alpha$ | 5–65° |
| Average buried depth | $h$ | 2–8 m |
| Distance between surface water level and the top of the slope | $L_1$ | 16–19 m |
| Distance between groundwater level and the top of the slope | $L_2$ | 0–6 m |

Table 7 lists the results of the orthogonal experiment of sand slope stability under water level fluctuation. The factors of safety of trials 27, 28, 31, and 32 are negative; however, according to the principle of model derivation, under these four sets of parameters, the anchoring force provided by the umbrella anchor is greater than the sliding force of the slope element, and the value is negative. In practice, the slope is still stable at this time.

**Table 7.** Results of the orthogonal experiment of sand slope stability under water level fluctuation.

| | 1 | 2 | 3 | 4 | 5 | |
|---|---|---|---|---|---|---|
| Trial No. | $n$ (Row) | $\alpha$ (°) | $h$ (m) | $L_1$ (m) | $L_2$ (m) | $F$ |
| 1 | 1 | 5 | 2 | 19 | 0 | 0.46 |
| 2 | 1 | 5 | 4 | 18 | 2 | 0.84 |
| 3 | 1 | 5 | 6 | 17 | 4 | 1.25 |
| 4 | 1 | 5 | 8 | 16 | 6 | 1.66 |
| 5 | 1 | 25 | 2 | 19 | 2 | 0.66 |
| 6 | 1 | 25 | 4 | 18 | 0 | 0.81 |
| 7 | 1 | 25 | 6 | 17 | 6 | 1.92 |
| 8 | 1 | 25 | 8 | 16 | 4 | 2.57 |
| 9 | 1 | 45 | 2 | 18 | 4 | 0.85 |
| 10 | 1 | 45 | 4 | 19 | 6 | 1.57 |
| 11 | 1 | 45 | 6 | 16 | 0 | 1.84 |
| 12 | 1 | 45 | 8 | 17 | 2 | 5.39 |
| 13 | 1 | 65 | 2 | 18 | 6 | 1.00 |
| 14 | 1 | 65 | 4 | 19 | 4 | 1.46 |
| 15 | 1 | 65 | 6 | 16 | 2 | 3.30 |
| 16 | 1 | 65 | 8 | 17 | 0 | 12.44 |
| 17 | 2 | 5 | 2 | 16 | 0 | 0.68 |
| 18 | 2 | 5 | 4 | 17 | 2 | 1.35 |
| 19 | 2 | 5 | 6 | 18 | 4 | 2.05 |
| 20 | 2 | 5 | 8 | 19 | 6 | 2.81 |
| 21 | 2 | 25 | 2 | 16 | 2 | 1.05 |
| 22 | 2 | 25 | 4 | 17 | 0 | 1.90 |
| 23 | 2 | 25 | 6 | 18 | 6 | 6.44 |

**Table 7.** *Cont.*

| | 1 | 2 | 3 | 4 | 5 | |
|---|---|---|---|---|---|---|
| Trial No. | $n$ (Row) | $\alpha$ (°) | $h$ (m) | $L_1$ (m) | $L_2$ (m) | $F$ |
| 24 | 2 | 25 | 8 | 19 | 4 | 60.12 |
| 25 | 2 | 45 | 2 | 17 | 4 | 1.41 |
| 26 | 2 | 45 | 4 | 16 | 6 | 7.21 |
| 27 | 2 | 45 | 6 | 19 | 0 | −5.13 |
| 28 | 2 | 45 | 8 | 18 | 2 | −2.47 |
| 29 | 2 | 65 | 2 | 17 | 6 | 1.76 |
| 30 | 2 | 65 | 4 | 16 | 4 | 15.60 |
| 31 | 2 | 65 | 6 | 19 | 2 | −1.63 |
| 32 | 2 | 65 | 8 | 18 | 0 | −0.86 |

Table 8 shows the results of the range analysis, and Figure 7 shows the response graph of the relationship among various factors and stability. The order of sensitivity of each factor is $n > h > L_2 > \alpha > L_1$.

**Table 8.** Range analysis of various factors.

| Factor | 1 | 2 | 3 | 4 | 5 |
|---|---|---|---|---|---|
| | $n$ | $\alpha$ | $h$ | $L_1$ | $L_2$ |
| $K_{1j}$ | 1.70 | 1.39 | 0.98 | 1.39 | 1.14 |
| $K_{2j}$ | 2.66 | 2.19 | 2.16 | 2.00 | 2.10 |
| $K_{3j}$ | | 3.04 | 2.80 | 2.14 | 1.60 |
| $K_{4j}$ | | 1.88 | 3.11 | 2.61 | 3.05 |
| $R_j$ | 2.18 | 1.65 | 2.12 | 1.23 | 1.91 |
| Rank | 1 | 4 | 2 | 5 | 3 |

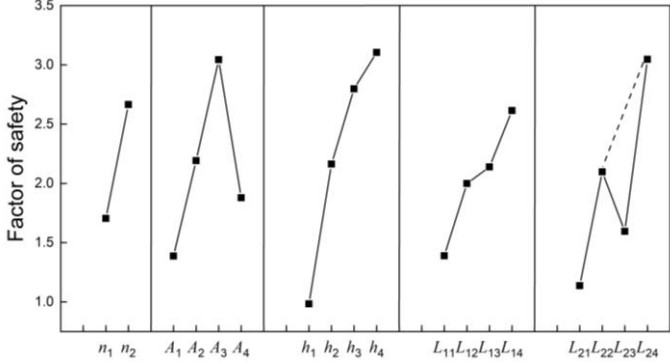

**Figure 7.** Response graph for the main effects according to Table 8.

Table 9 shows the results of the analysis of variance; the order of the analysis of variance is as follows: $n > h > \alpha > L_2 > L_1$. The $p$-value of $n$ is 0.043, less than 0.05. This implies that the layout density of the umbrella anchors $n$ has a more significant influence on slope stability.

**Table 9.** Variance analysis.

| Source | Sum Sq. | Mean Sq. | $F$ | Prob > $F$ | Rank |
|--------|---------|----------|-----|------------|------|
| $n$ | 11.32 | 11.32 | 5.21 | 0.043 | 1 |
| $\alpha$ | 15.21 | 5.07 | 2.33 | 0.130 | 3 |
| $h$ | 19.60 | 6.53 | 3.00 | 0.077 | 2 |
| $L_1$ | 1.62 | 0.54 | 0.25 | 0.861 | 5 |
| $L_2$ | 4.67 | 1.56 | 0.72 | 0.563 | 4 |
| Error | 23.92 | 2.17 | | | |
| Total | 74.66 | | | | |

*3.2. Sensitivity Analysis of Anchored Clay Slope*

Case 2 analyzes the sensitivity of a clay slope. Figure 8 shows the profile and a simplified model of the Cangjiang bridge–Yingpan slope. It can be divided into three secondary slopes, where part I is critical stable, part II has local deformation in the shallow layer, part III is relatively stable compared with parts I and II. The bedrock is the Jurassic Bazhulu Formation ($J_3b$). Table 10 shows the relevant basic parameters in the sensitivity analysis, including the internal friction angle, soil weight and thickness of sliding mass, and effective cohesion force. At the same time, the water head of the slope and slope inclination is also considered.

$$F = \frac{\frac{mc'}{\gamma z} + \left( \cos^2 \beta - \frac{\gamma_w lh}{\gamma z} \right) \tan n\varphi'}{\sin \beta \cos \beta} \tag{10}$$

where $F$ is the factor of safety, $mc'$ ($m > 1$) is the effective cohesion force of the anchoring system (kPa), $\gamma$ is the unit weight of sliding mass (kN/m$^3$), and $\beta$ is the slope inclination (°). Umbrella anchors can form an anchoring system in the slope, which influences the properties of the soil. The effective cohesion force ($c$) and effective internal friction angle ($\varphi$) could be increased by the anchoring system, and the pore water pressure ($h$) decreases. Therefore, $c$, $\varphi$, and $h$ are described by adding coefficients $l$, $m$, and $n$ ($m > 1$, $n > 1$, $l < 1$), respectively, after the anchoring system is formed; $mc'$ ($m > 1$) is the effective cohesion force of the anchoring system (kPa), $n\varphi'$ ($n > 1$) is the effective internal friction angle of the anchoring system (°), and $lh$ ($l < 1$) is the water head height of the piezometer after the anchorage system is formed (m). [57]. Table 11 shows the assumed values of $m$, $n$, and $l$.

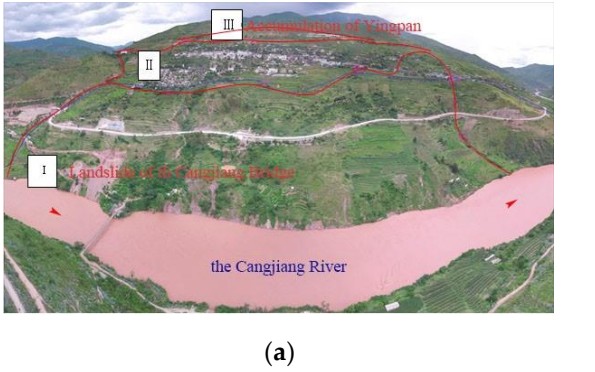

(**a**)

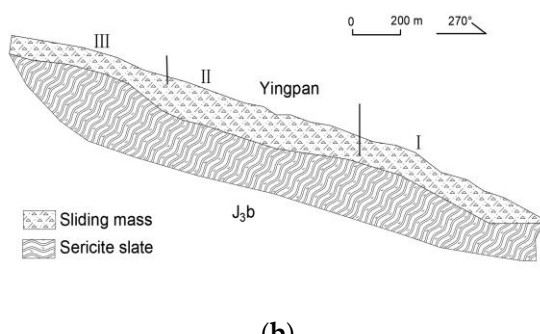

(**b**)

**Figure 8.** The Cangjiang bridge–Yingpan slope and geological section: (**a**) a photo, (**b**) a cross-section.

**Table 10.** Related parameters in the study area.

| Mass | $\gamma$ (kN/m$^3$) | $\varphi$ (°) | $c$ (kPa) |
|------|---------------------|---------------|-----------|
| Sliding mass | 20–21 | 20–24 | 20–40 |
| Soil mass in sliding zone | 18–20 | 19–23 | 15–20 |

**Table 11.** Coefficients of the anchorage system.

|       | $m$ | $n$ | $l$ |
|-------|-----|-----|-----|
| $F$   | 1   | 1   | 1   |
| $F_1$ | 1.1 | 1.1 | 0.9 |
| $F_2$ | 1.1 | 1.2 | 0.9 |
| $F_3$ | 1.2 | 1.2 | 0.8 |
| $F_4$ | 1.2 | 1.1 | 0.8 |

Figure 9 shows different *F*-values with different values of *m*, *n*, and *l*, indicating a similar trend. The trial results (Table 12) indicate that the factor sensitivity of the study area can be analyzed by assuming the values of *m*, *n*, and *l*.

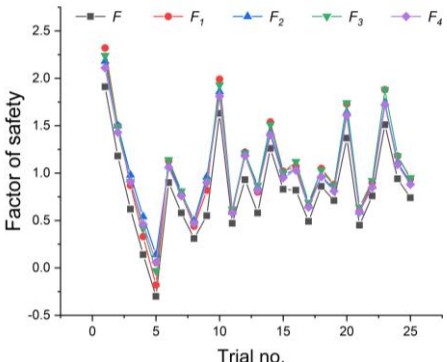

**Figure 9.** Comparison of the trend of the factor of safety.

**Table 12.** Results of the orthogonal experiment of the Cangjiang bridge–Yingpan slope's stability under water level fluctuations.

| Trial No. | $z$ (m) | $c$ (kPa) | $h$ (m) | $\beta$ (°) | $\varphi$ (°) | $\gamma$ (kN/m³) | $F$   | $F_1$ | $F_2$ | $F_3$ | $F_4$ |
|-----------|---------|-----------|---------|-------------|---------------|------------------|-------|-------|-------|-------|-------|
| 1  | 5  | 20 | 0  | 17 | 20 | 20.00 | 1.91  | 2.32  | 2.18 | 2.24  | 2.11 |
| 2  | 5  | 25 | 5  | 21 | 21 | 20.25 | 1.18  | 1.50  | 1.50 | 1.49  | 1.43 |
| 3  | 5  | 30 | 10 | 25 | 22 | 20.50 | 0.62  | 0.87  | 0.98 | 0.91  | 0.91 |
| 4  | 5  | 35 | 15 | 29 | 23 | 20.75 | 0.14  | 0.33  | 0.54 | 0.42  | 0.46 |
| 5  | 5  | 40 | 20 | 33 | 24 | 21.00 | −0.30 | −0.18 | 0.14 | −0.03 | 0.06 |
| 6  | 10 | 20 | 5  | 25 | 23 | 21.00 | 0.90  | 1.13  | 1.08 | 1.14  | 1.06 |
| 7  | 10 | 25 | 10 | 29 | 24 | 20.00 | 0.58  | 0.77  | 0.79 | 0.81  | 0.76 |
| 8  | 10 | 30 | 15 | 33 | 20 | 20.25 | 0.31  | 0.44  | 0.50 | 0.47  | 0.47 |
| 9  | 10 | 35 | 20 | 17 | 21 | 20.50 | 0.55  | 0.82  | 0.96 | 0.92  | 0.90 |
| 10 | 10 | 40 | 0  | 21 | 22 | 20.75 | 1.63  | 1.99  | 1.86 | 1.93  | 1.81 |
| 11 | 15 | 20 | 10 | 33 | 21 | 20.75 | 0.47  | 0.60  | 0.59 | 0.62  | 0.58 |
| 12 | 15 | 25 | 15 | 17 | 22 | 21.00 | 0.93  | 1.22  | 1.21 | 1.21  | 1.18 |
| 13 | 15 | 30 | 20 | 21 | 23 | 20.00 | 0.58  | 0.80  | 0.85 | 0.87  | 0.82 |
| 14 | 15 | 35 | 0  | 25 | 24 | 20.25 | 1.26  | 1.54  | 1.43 | 1.51  | 1.40 |
| 15 | 15 | 40 | 5  | 29 | 20 | 20.50 | 0.83  | 1.02  | 0.98 | 1.01  | 0.95 |
| 16 | 20 | 20 | 15 | 21 | 24 | 20.50 | 0.82  | 1.07  | 1.04 | 1.12  | 1.03 |
| 17 | 20 | 25 | 20 | 25 | 20 | 20.75 | 0.49  | 0.65  | 0.66 | 0.69  | 0.64 |
| 18 | 20 | 30 | 0  | 29 | 21 | 21.00 | 0.86  | 1.05  | 0.97 | 1.03  | 0.96 |
| 19 | 20 | 35 | 5  | 33 | 22 | 20.00 | 0.71  | 0.88  | 0.83 | 0.87  | 0.81 |
| 20 | 20 | 40 | 10 | 17 | 23 | 20.25 | 1.37  | 1.73  | 1.64 | 1.74  | 1.61 |
| 21 | 25 | 20 | 20 | 29 | 22 | 20.25 | 0.45  | 0.600 | 0.59 | 0.64  | 0.59 |
| 22 | 25 | 25 | 0  | 33 | 23 | 20.50 | 0.76  | 0.9   | 0.86 | 0.92  | 0.85 |
| 23 | 25 | 30 | 5  | 17 | 24 | 20.75 | 1.51  | 1.88  | 1.74 | 1.88  | 1.72 |
| 24 | 25 | 35 | 10 | 21 | 20 | 21.00 | 0.94  | 1.18  | 1.11 | 1.18  | 1.09 |
| 25 | 25 | 40 | 15 | 25 | 21 | 20.00 | 0.74  | 0.94  | 0.90 | 0.95  | 0.88 |

Figure 10 shows that the stability of the slope in the study area is positively correlated with the effective cohesion force of the sliding mass and the internal friction angle and negatively correlated with the slope inclination, water head height, soil weight, and thickness of the sliding mass.

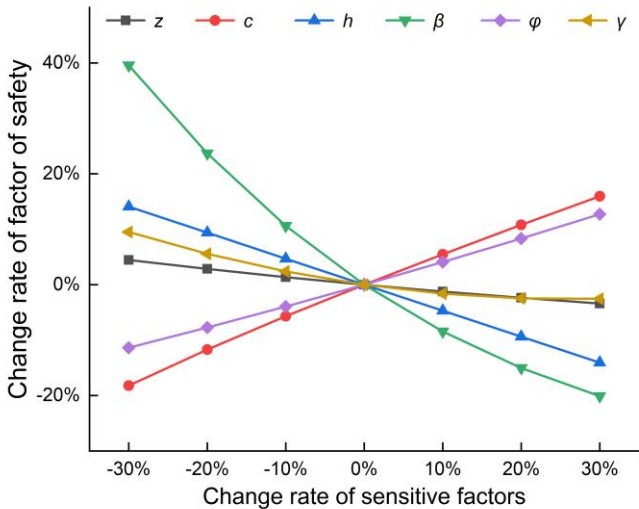

**Figure 10.** Comprehensive graph of the single factor sensitivity analysis.

The results of the range analysis (Table 13) show that the sensitivity of each factor is ranked as follows: $h > \beta > c > z > \varphi > \gamma$. Figure 11 shows the trend of the influence of the main factors on the stability of the Cangjiang bridge–Yingpan slope. Table 14 shows that the *p*-value of the water head height and the slope inclination in the study area is less than 0.01, which indicates it has a prominent impact on the factor of safety in the study area. The *p*-value of the soil weight of the sliding mass is 0.035, less than 0.05, which means that the soil weight of the sliding mass has a more significant impact on the factor of safety.

**Table 13.** Range analysis for main effects on the stability of the Cangjiang bridge–Yingpan slope under water level fluctuations.

| Factor | 1 | 2 | 3 | 4 | 5 | 6 |
|---|---|---|---|---|---|---|
| | $z$ | $c$ | $h$ | $\beta$ | $\varphi$ | $\gamma$ |
| $K_{1j}$ | 1.18 | 1.04 | 1.74 | 1.47 | 1.16 | 1.19 |
| $K_{2j}$ | 1.09 | 1.18 | 1.18 | 1.18 | 1.18 | 1.18 |
| $K_{3j}$ | 1.06 | 1.33 | 0.63 | 0.98 | 1.21 | 1.18 |
| $K_{4j}$ | 1.05 | 1.48 | 0.07 | 0.83 | 1.23 | 1.18 |
| $K_{5j}$ | 1.04 | 1.63 | - | 0.72 | 1.25 | 1.17 |
| $R_j$ | 0.14 | 0.59 | 1.67 | 0.75 | 0.09 | 0.02 |
| Rank | 4 | 3 | 1 | 2 | 5 | 6 |

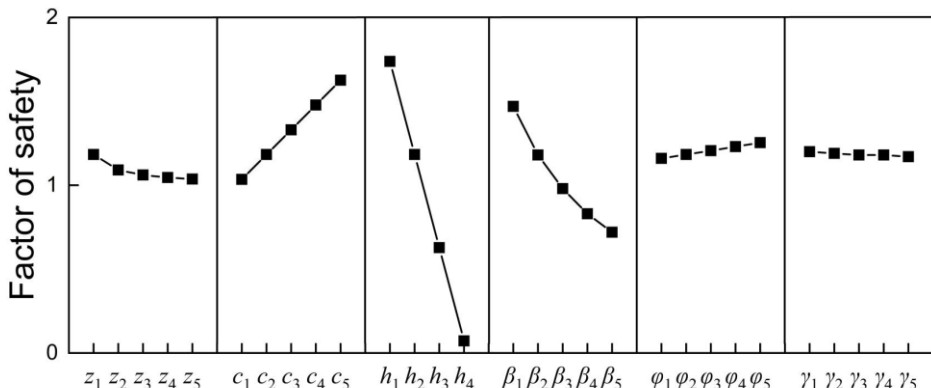

**Figure 11.** Response graph for the main effects according to Table 13.

**Table 14.** Results of variance analysis of the Cangjiang bridge–Yingpan slope's stability.

| Source | Sum Sq. | d.f | Mean Sq. | F | Prob > F |
|--------|---------|-----|----------|---|----------|
| $c$ | 0.15 | 4 | 0.036 | 4.52 | 0.087 |
| $h$ | 3.04 | 4 | 0.760 | 93.43 | 0.001 |
| $\beta$ | 3.40 | 4 | 0.849 | 104.39 | 0.001 |
| $\varphi$ | 0.09 | 4 | 0.022 | 2.65 | 0.184 |
| $\gamma$ | 0.26 | 4 | 0.064 | 7.88 | 0.035 |
| Error | 0.03 | 4 | 0.008 | | |
| Total | 6.96 | 24 | 0 | | |

## 4. Summary and Conclusions

The fluctuations of the reservoir water level play a pivotal role in inducing slope failures [53]. In this paper, the formula of the factor of safety of sand slopes under water level fluctuation was improved by adding anchoring factors. Then, the sensitivity of anchored sand and clay slopes under water level fluctuations was analyzed. A sensitivity order of related factors and the corresponding main control factors of the safety of slopes was obtained through an ISSSA model. Case 2 considered that the anchoring system of the umbrella anchor is formed with the surrounding cementation, and the effect of the umbrella anchor is directly substituted as the enhancement coefficient ($m$, $n$, $l$), which is different to Case 1 to some extent, so the sensitivity order of a few factors is different. The order showed that the main controlling factors for the stability of the anchored sand slope under the fluctuation of the water level are the slope inclination and the layout density of the anchors (the number of umbrella anchors arranged in the vertical direction of the slope within the unit width). The water head height and slope inclination are the main controlling factors for the stability of the Cangjiang bridge–Yingpan slope.

With an increase in the anchorage angle $\alpha$, the factor of safety will increase at first and then decrease. In other words, there is an optimum anchorage angle that has the greatest influence on slope stability. The angle should be in the range of 25–45 degrees.

The slope will be more stable when the factors of surface and groundwater level change synchronously. If a single water level factor changes suddenly in a short time, this will easily lead to slope instability, and a sudden drop in water level is more likely to cause a landslide. If the surface water level is less than 10% of the slope height and the surface water level rises sharply, due to the influence of hysteresis, the factor of safety will first increase and then decrease. When the surface water level is at 20%~30% of the slope height, the fluctuation of water level has the greatest influence on the factor of safety. When the water level factor fluctuates greatly, we should pay attention to the slope situation and implement corresponding measures in a timely manner in order to prevent the occurrence of landslide disasters.

There are some limitations in this work; for instance, the discussion on the influence of water level fluctuation factors on the factor of safety is mainly based on sand slopes.

Moreover, the comprehensive influence of the fluctuation of the surface and groundwater level factors on the factor of safety should be analyzed for the clay slope. Due to the limitation of the model conditions, the sensitivity analysis did not consider the influence of factors such as the speed of water level changes and the groundwater seepage channels on slope stability.

**Author Contributions:** Conceptualization, J.L. and W.S.; methodology, J.L. and W.S.; software, J.L.; validation, J.L. and W.S.; formal analysis, J.L. and W.S.; investigation, J.L.; resources, J.L. and W.S.; data curation, J.L. and W.S.; writing—original draft preparation, J.L.; writing—review and editing, J.L. and W.S.; visualization, J.L.; supervision, W.S.; project administration, W.S.; funding acquisition, W.S. All authors have read and agreed to the published version of the manuscript.

**Funding:** This research was funded by the National Key Research and Development Plan of China (No. 2017YFC1501303) and the Postgraduate Research & Practice Innovation Program of Jiangsu Province (Grant KYCX21_2317).

**Institutional Review Board Statement:** Not applicable.

**Informed Consent Statement:** Not applicable.

**Data Availability Statement:** Not applicable.

**Conflicts of Interest:** The authors declare no conflict of interest.

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
