# Peer review of "Sensitivity Analysis of Anchored Slopes under Water Level Fluctuations: A Case Study of Cangjiang Bridge—Yingpan Slope in China"

_applsci, doi:10.3390/app11157137_

Round 1

Reviewer 1 Report

Dear Authors,

The work contains important considerations on maintaining the safety of slopes by using umbrella anchores. The work contains a few errors which do not detract from its substantive value. I also advise the authors to make a linguistic proofreading. In my opinion, after taking into account my suggestions and comments, the work is ready for publication.

Reviewer 2 Report

The article presents a sensitivity slope stability analysis of a reinforced slope through the classical limit equilibrium method. The manuscript could be written in a more formal and scientific way; besides English needs improvement. Although the paper concerns anchored slopes, nothing is said concerning this stabilization technique. Furthermore, some issues are found throughout the paper. Therefore, according to this Reviewer, a major revision would be necessary before the paper can be further considered for possible publication in Applied Sciences. All details are summed up in the following.

Required changes:

  1. Some language editing is needed, both from English and scientific point of view.
  2. Introduction is completely missing of any trace of explanation about slope stabilization techniques by means of anchors. In this context, a detailed explanation should be added. Furthermore, a comparison between this stabilization technique and other alternatives is required, such as for example drainage systems [1] and stabilizing piles [2]. For the sake of completeness, these suggested references are reported at the end of this report.
  3. Lines 32-35: this concept is very important and should be deepened. Some case studies could be mentioned, such as the following ones [3, 4].
  4. Equation 1 is not of general validity. Rather, it is valid only for cohesionless and dry slopes, characterized by a plane slip surface. This should be pointed out in the text.
  5. Equations 3/4/5: it is not clear the geometry that these equations refer to. A scheme should be added in the concerning section.
  6. Table 1: add the value of the friction angle. Besides, it is not clear what the coefficient ξ is, nor its physical meaning.
  7. Lines 237-243: it is not clear what the letter l, m, n

SUGGESTED REFERENCES

[1] Wei, Zl., Wang, Df., Xu, Hd. et al. Clarifying the effectiveness of drainage tunnels in landslide controls based on high-frequency in-site monitoring. Bull Eng Geol Environ 79, 3289–3305 (2020).

[2] Troncone A., Pugliese L., Lamanna G., Conte E. (2021). Prediction of rainfall-induced landslide movements in the presence of stabilizing piles. Engineering Geology 288, 106143.

[3] Wei ZL, Lü Q, Sun HY, Shang YQ (2019) Estimating the rainfall threshold of a deep-seated landslide by integrating models for predicting the groundwater level and stability analysis of the slope. Eng Geol 253:14–26

[4] Troncone, A.; Pugliese, L.; Conte, E. Run-Out Simulation of a Landslide Triggered by an Increase in the Groundwater Level Using the Material Point Method. Water 2020, 12, 2817.

Round 2

Reviewer 2 Report

All the issues raised during the previous round of review have been properly addressed. The manuscript has been improved a lot, both from scientific and language point of view. According to this reviewer, it can be accepted in the present form.
